# Learning Vision-Guided Quadrupedal Locomotion End-to-End with Cross-Modal Transformers

**Ruihan Yang**[*]        **Minghao Zhang**[*]        **Nicklas Hansen**        **Huazhe Xu**        **Xiaolong Wang**
UC San Diego        Tsinghua University        UC San Diego        UC Berkeley        UC San Diego

## Abstract

We propose to address quadrupedal locomotion tasks using Reinforcement Learning (RL) with a Transformer-based model that learns to combine proprioceptive information and high-dimensional depth sensor inputs. While learning-based locomotion has made great advances using RL, most methods still rely on domain randomization for training blind agents that generalize to challenging terrains. Our key insight is that proprioceptive states only offer contact measurements for immediate reaction, whereas an agent equipped with visual sensory observations can learn to proactively maneuver environments with obstacles and uneven terrain by anticipating changes in the environment many steps ahead. In this paper, we introduce *LocoTransformer*, an end-to-end RL method that leverages both proprioceptive states and visual observations for locomotion control. We evaluate our method in challenging simulated environments with different obstacles and uneven terrain. We transfer our learned policy from simulation to a real robot by running it indoors and in the wild with unseen obstacles and terrain. Our method not only significantly improves over baselines, but also achieves far better generalization performance, especially when transferred to the real robot. Our project page with videos is at https://rchalyang.github.io/LocoTransformer/.

## 1 Introduction

Legged locomotion is one of the core problems in robotics research. It expands the reach of robots and enables them to solve a wide range of tasks, from daily life delivery to planetary exploration in challenging, uneven terrain (Delcomyn & Nelson, 2000; Arena et al., 2006). Recently, besides the success of Deep Reinforcement Learning (RL) in navigation (Mirowski et al., 2017; Gupta et al., 2019; Yang et al., 2019; Kahn et al., 2021) and robotic manipulation (Levine et al., 2018; 2016; Tian et al., 2019; Jain et al., 2019b), we have also witnessed the tremendous improvement of locomotion skills for quadruped robots, allowing them to walk on uneven terrain (Xie et al., 2020; 2021), and even generalize to real-world with mud, snow, and running water (Lee et al., 2020a).

While these results are encouraging, most RL approaches focus on learning a robust controller for *blind* quadrupedal locomotion, using only the proprioceptive state. For example, Lee et al. (2020a) utilize RL with domain randomization and large-scale training in simulation to learn a robust quadrupedal locomotion policy, which can be applied to challenging terrains. However, is domain randomization with blind agents really sufficient for general legged locomotion?

By studying eye movement during human locomotion, Matthis et al. (2018) show that humans rely heavily on eye-body coordination when walking and that the gaze changes depending on characteristics of the environment, e.g., whether humans walk in flat or rough terrain. This finding motivates the use of visual sensory input to improve quadrupedal locomotion on uneven terrain. While handling uneven terrain is still possible without the vision, a blind agent is unable to consistently avoid large obstacles as shown in Figure 1. To maneuver around such obstacles, the agent needs to perceive the obstacles at a distance and dynamically make adjustments to its trajectory to avoid any collision. Likewise, an agent navigating rough terrain (mountain and forest in Figure 1) may also

---

[*]Equal Contribution

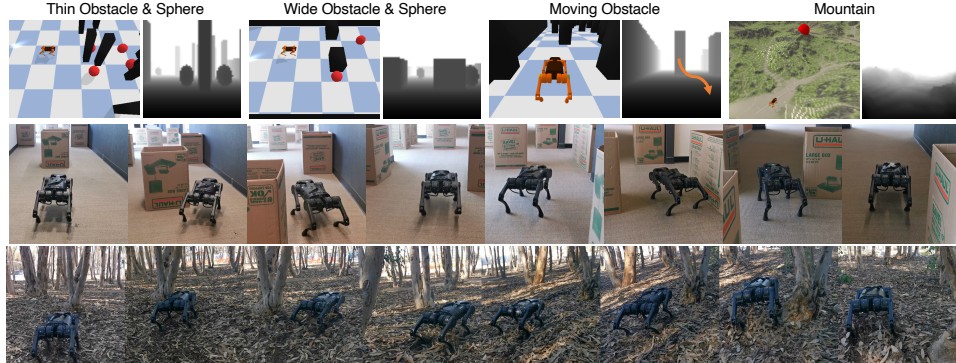

Figure 1: **Overview of simulated environments & real robot trajectories.** *Top row* shows the simulated environments. For each sample, the left image is the environment and the right image is the corresponding observation. Agents are tasked to move forward while avoiding black obstacles and collecting red spheres. *Following two rows* show the deployment of the RL policy to a real robot in an indoor hallway with boxes and a forest with trees. Our robot successfully utilizes the visual information to traverse the complex environments.

benefit from vision by anticipating changes in the terrain before contact, and visual observations can therefore play an important role in improving locomotion skills.

In this paper, we propose to combine proprioceptive states and first-person-view visual inputs with a cross-modal Transformer for learning locomotion RL policies. Our key insight is that proprioceptive states (i.e. robot pose, Inertial Measurement Unit (IMU) readings, and local joint rotations) provide a precise measurement of the current robot status for *immediate* reaction, while visual inputs from a depth camera help the agent plan to maneuver uneven terrain and large obstacles in the path. We fuse the proprioceptive state and depth image inputs using Transformers (Vaswani et al., 2017; Tsai et al., 2019) for RL, which enables the model to reason with complementary information from both modalities. Additionally, Transformers also offer a mechanism for agents to attend to specific visual regions (e.g. objects or uneven ground) that are critical for their long-term and short-term decision making, which may in turn lead to a more generalizable and interpretable policy.

Our Transformer-based model for locomotion, *LocoTransformer*, consists of two encoders for inputs (an MLP for proprioceptive states, a ConvNet for depth image inputs) and Transformer encoders for multi-modal fusion. We obtain a feature embedding from the proprioceptive states and multiple image patch embeddings from the depth images, which are used jointly as token inputs for the Transformer encoders. Feature embeddings for both modalities are then updated with information propagation among all the tokens using self-attention. We combine both features for action prediction. The model is trained end-to-end without hierarchical RL (Peng et al., 2017; Jiang et al., 2019; Jain et al., 2019a) nor pre-defined controllers (Da et al., 2020; Escontrela et al., 2020).

We experiment on both simulated and real environments as shown in Figure 1. Our tasks in simulation include maneuvering around obstacles of different sizes, dynamically moving obstacles, and rough mountainous terrain. With simulation-to-real (sim2real) transfer, we deploy the policies to the robot on indoor hallways with box obstacles and outdoor forests with trees and uneven terrain. We show that learning policies with both proprioceptive states and vision significantly improve locomotion control, and the policies further benefit from adopting cross-modal Transformer. We also show that *LocoTransformer* generalizes much better to unseen environments, especially for sim2real transfer. We highlight our main contributions as follows:

- Going beyond blind robots, we introduce visual information into end-to-end RL policies for quadrupedal locomotion to traverse complex terrain with different kinds of obstacles.
- We propose *LocoTransformer*, which fuses proprioceptive states and visual inputs for better multi-modal reasoning in sequential decision making.
- To the best of our knowledge, this is the first work which deploys vision-based RL policy on running real quadrupedal robot avoiding obstacles and trees in the wild.

## 2 RELATED WORK

**Learning Legged Locomotion.** Developing legged locomotion controllers has been a long-standing problem in robotics (Miura & Shimoyama, 1984; Raibert, 1984; Torkos & van de Panne, 1998; Geyer

et al., 2003; Yin et al., 2007; Bledt et al., 2018). While encouraging results have been achieved using Model Predictive Control (MPC) and trajectory optimization (Gehring et al., 2013; Carlo et al., 2018; Di Carlo et al., 2018; Carius et al., 2019; Ding et al., 2019; Grandia et al., 2019; Bledt & Kim, 2020; Sun et al., 2021), these methods require in-depth knowledge of the environment and substantial manual parameter tuning, which makes these methods challenging to apply to complex environments. Alternatively, model-free RL can learn general policies on challenging terrain (Kohl & Stone, 2004; Zhang et al., 2018; Luo et al., 2020; Peng et al., 2018; Tan et al., 2018; Hwangbo et al., 2019; Lee et al., 2020a; Iscen et al., 2018; Jain et al., 2019a; Xie et al., 2021; Kumar et al., 2021). Xie et al. (2020) use dynamics randomization to generalize RL locomotion policy in different environments, and Peng et al. (2020) use animal videos to provide demonstrations for imitation learning. However, most approaches currently rely only on proprioceptive states without other visual inputs. In this work, we propose to incorporate both visual and proprioceptive inputs using a Transformer for RL policy, which allows the quadruped robot to simultaneously move and plan its trajectory.

**Vision-based Reinforcement Learning.** To generalize RL to real-world applications beyond state inputs, a lot of effort has been made in RL with visual inputs (Sax et al., 2018; Jaderberg et al., 2017; Levine et al., 2016; 2018; Pathak et al., 2017; Jain et al., 2019b; Mnih et al., 2015a; Lin et al., 2019; Yarats et al., 2019; Laskin et al., 2020; Stooke et al., 2020; Schwarzer et al., 2020). For example, Srinivas et al. (2020) propose to apply contrastive self-supervised representation learning (He et al., 2020) with the RL objective in vision-based RL. Hansen & Wang (2021) further extend the joint representation learning and RL for better generalization to out-of-distribution environments. Researchers have also looked into combining multi-modalities with RL for manipulation tasks (Lee et al., 2020b; Calandra et al., 2018) and locomotion control (Heess et al., 2017; Merel et al., 2020). Escontrela et al. (2020) propose to combine proprioceptive states and LiDAR inputs for learning quadrupedal locomotion with RL using MLPs. Jain et al. (2020) propose to use Hierarchical RL (HRL) for locomotion, which learns high-level policies under visual guidance and low-level motor control policies with IMU inputs. Different from previous work, we provide a simple approach to combine proprioceptive states and visual inputs with a Transformer model in an end-to-end manner without HRL. Our LocoTransformer not only performs better in challenging environments but also achieves better generalization results in unseen environments and with the real robot.

**Transformers and Multi-modal Learning.** The Transformer model has been widely applied in the fields of language processing (Vaswani et al., 2017; Devlin et al., 2018; Brown et al., 2020) and visual recognition and synthesis (Wang et al., 2018; Parmar et al., 2018; Child et al., 2019; Dosovitskiy et al., 2020; Carion et al., 2020; Chen et al., 2020a). Besides achieving impressive performance in a variety of language and vision tasks, the Transformer also provides an effective mechanism for multi-modal reasoning by taking different modality inputs as tokens for self-attention (Su et al., 2019; Tan & Bansal, 2019; Li et al., 2019; Sun et al., 2019; Chen et al., 2020b; Li et al., 2020; Prakash et al., 2021; Huang et al., 2021; Hu & Singh, 2021; Akbari et al., 2021; Hendricks et al., 2021). For example, Sun et al. (2019) propose to use a Transformer to jointly model video frames and their corresponding captions from instructional videos for representation learning. Going beyond language and vision, we propose to utilize cross-modal Transformers to fuse proprioceptive states and visual inputs. To our knowledge, this is the first work using cross-modal Transformers for locomotion.

## 3 REINFORCEMENT LEARNING BACKGROUND

We model the interaction between the robot and the environment as an MDP (Bellman, 1957) $(S, A, P, \mathcal{R}, H, \gamma)$, where $s \in S$ are states, $a \in A$ are actions, $P(s'|s, a)$ is transition function, $\mathcal{R}$ is reward function, $H$ is finite episode horizon, and $\gamma$ is discount factor. The Agent learn a policy $\pi_\theta$ parameterized by $\theta$ to output actions distribution conditioned on current state. The goal of agent is to learn $\theta$ that maximizes the discounted episode return: $R = \mathbb{E}_{\tau \sim p_\theta(\tau)}[\sum_{t=0}^{H} \gamma^t r_t]$, where $r_t \sim \mathcal{R}(s_t, a_t)$ is the reward for time step $t$, $\tau \sim p_\theta(\tau)$ is the trajectory.

## 4 METHOD

We propose to incorporate both proprioceptive and visual information for locomotion tasks using a novel Transformer model, *LocoTransformer*. Figure 2 provides an overview of our architecture. Our model consists of the following two components: (i) Separate modality encoders for proprioceptive and visual inputs that project both modalities into a latent feature space; (ii) A shared Transformer encoder that performs cross-modality attention over proprioceptive features and visual features, as well as spatial attention over visual tokens to predict actions and predict values.

## 4.1 SEPARATE MODALITY ENCODERS

In our setting, the agent utilizes both proprioceptive states and visual observations for decision-making. Proprioceptive state and visual observation are distinctively different modalities: the proprioceptive input is a 93-D vector, and we use stacked first-person view depth images to encode the visual observations. To facilitate domain-specific characteristics of both modalities, we use two separate, domain-specific encoders for proprioceptive and visual data respectively, and unify the representation in a latent space. We now introduce the architectural design of each encoder, and how features are converted into tokens for the Transformer encoder.

We use an MLP to encode the proprioceptive input into proprioceptive features $E^{\text{prop}} \in \mathbb{R}^{C^{\text{prop}}}$, where $C^{\text{prop}}$ is the proprioceptive feature dimension.

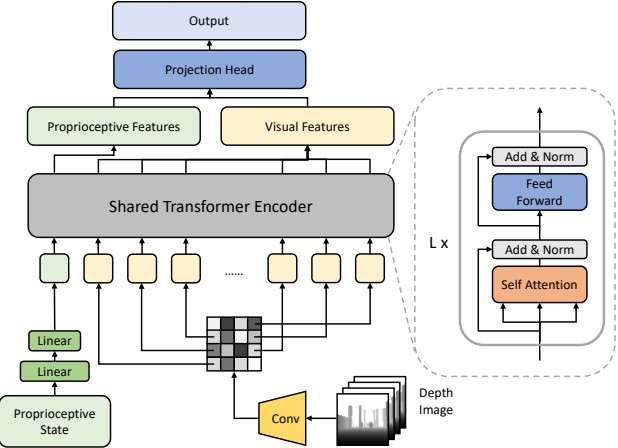

Figure 2: **Network Architecture**. We process proprioceptive states with a MLP and depth images with a ConvNet. We take proprioceptive embedding as a single token, split the spatial visual feature representation into $N \times N$ tokens and feed all tokens into the Transformer encoder. The output tokens are further processed by the projection head to predict value or action distribution.

We encode additionally provided visual information using a ConvNet. The ConvNet encoder forwards the stacked depth image inputs into a spatial representation $E^{\text{visual}}$ with shape $C \times N \times N$, where $C$ is the channel number, and $N$ is the width and height of the representation. The depth images are from the first-person view from the frontal of the robot, which captures the obstacles and terrain from the perspective of the acting robot. However, for first-person view, the moving camera and limited field-of-view make learning visual policies significantly more challenging. For instance, changes in robot pose can result in changes in visual observations. This makes it essential to leverage proprioceptive information to improve visual understanding. In the following, we present our proposed method for fusing the two modalities and improving their joint representation using a Transformer.

## 4.2 TRANSFORMER ENCODER

We introduce the Transformer encoder to fuse the visual observations and the proprioceptive states for decision making. Given a spatial visual features with shape $C \times N \times N$ from the ConvNet encoder, we split the spatial features into $N \times N$ different $C$-dimensional token embeddings $t^{\text{visual}} \in \mathbb{R}^C$ (illustrated as yellow tokens in Figure 2), each corresponding to a local visual region. We use a linear layer to project the proprioceptive features into a $C$-dimensional token embedding $t^{\text{prop}} \in \mathbb{R}^C$ (illustrated as a green token in Figure 2). Formally, we have $N \times N + 1$ tokens in total obtained by:

$$t^{\text{prop}} = W^{\text{prop}}(E^{\text{prop}}) + b^{\text{prop}} \qquad\qquad t^{\text{prop}} \in \mathbb{R}^C \qquad (1)$$

$$T_0 = [t^{\text{prop}}, t^{\text{visual}}_{0,0}, t^{\text{visual}}_{0,1}, ..., t^{\text{visual}}_{N-1,N-1}] \qquad\qquad t^{\text{visual}}_{i,j} \in \mathbb{R}^C \qquad (2)$$

where $t^{\text{visual}}_{i,j}$ is the token at spatial position $(i, j)$ of the visual features $E^{\text{visual}}$, and $W^{\text{prop}}, b^{\text{prop}}$ are the weights and biases, respectively, of the linear projection for proprioceptive token embedding. In the following, we denote $T_m \in \mathbb{R}^{(N^2+1) \times C}$ as the sequence of tokens after $m$ Transformer encoder layers, and $T_0$ as the input token sequence from Eq. 2.

We adopt a stack of Transformer encoder layers (Vaswani et al., 2017) to fuse information from proprioceptive and visual tokens. Specifically, we formulate the Self-Attention (SA) mechanism of the Transformer encoder as a scaled dot-product attention mechanism, omitting subscripts for brevity:

$$T^q, T^k, T^v = TU^q, TU^k, TU^v \qquad\qquad U^q, U^k, U^v \in \mathbb{R}^{C \times C} \qquad (3)$$

$$W^{\text{sum}} = \text{Softmax}(T^q T^{k\top} / \sqrt{D}) \qquad\qquad W^{\text{sum}} \in \mathbb{R}^{(N^2+1) \times (N^2+1)} \qquad (4)$$

$$\text{SA}(T) = W^{\text{sum}} T^v U^{\text{SA}} \qquad\qquad U^{\text{SA}} \in \mathbb{R}^{C \times C} \qquad (5)$$

where $D$ is the dimension of the self-attention layer. The SA mechanism first applies separate linear transformations on each input token $T$ to produce embeddings $T^q, T^k, T^v$ as defined in Eq. 3. We then compute a weighted sum over input tokens, where the weight $W_{i,j}^{\text{sum}}$ for each token pair $(t_i, t_j)$ is computed as the dot-product of elements $t_i$ and $t_j$ scaled by $1/\sqrt{D}$ and normalized by a Softmax operation. After a matrix multiplication between weights $W^{\text{sum}}$ and values $T^v$, we forward the result to a linear layer with parameters $U^{\text{SA}}$ as in Eq. 5, and denote this as the output $\text{SA}(T)$.

Each Transformer encoder layer consists of a self-attention layer, two LayerNorm (LN) layers with residual connections, and a 2-layer MLP as shown in Figure 2 (right). This is formally expressed as,

$$T'_m = \text{LN}(\text{SA}(T_m) + T_m), \quad T_{m+1} = \text{LN}(\text{MLP}(T'_m) + T'_m), \quad T_m, T_{m+1} \in \mathbb{R}^{(N^2+1) \times C} \quad (6)$$

where $T'_m$ is the normalized SA. Because SA is computed across visual tokens and single proprioceptive token, proprioceptive information may gradually vanish in multi-layer Transformers; the residual connections make the propagation of proprioceptive information through the network easier.

We stack $L$ Transformer encoder layers. Performing multi-layer self-attention on proprioceptive and visual features enables our model to fuse tokens from both modalities at multiple levels of abstraction. Further, we emphasize that a Transformer-based fusion allows spatial reasoning, as each visual token has a separate regional receptive field, and self-attention, therefore, enables the agent to explicitly attend to relevant visual regions. For modality-level fusion, direct application of a pooling operation across all tokens would easily dilute proprioceptive information since the number of visual tokens far exceed that of the proprioceptive token. To balance information from both modalities, we first pool information separately for each modality, compute the mean of all tokens from the same modality to get a single feature vector. We then concatenate the feature vectors of both modalities and project the concatenated vector into a final output vector using an MLP, which we denote the *projection head*.

**Observation Space.** In all environments, the agent receives both proprioceptive states and visual input as follows: (i) **proprioceptive data**: a 93-D vector consists of IMU readings, local joint rotations, and actions taken by the agent for the last three time steps; and (ii) **visual data**: stacked the most recent 4 dense depth image of shape $64 \times 64$ from a depth camera mounted on the head of the robot, which provides the agent with both spatial and temporal visual information.

**Implementation Details.** For the proprioceptive encoder and the projection head, we use a 2-layer MLP with hidden dimensions $(256, 256)$. Our visual encoder encode visual inputs into $4 \times 4$ spatial feature maps with 128 channels, following the architecture in Mnih et al. (2015b). Our shared Transformer consists of 2 Transformer encoder layers, each with a hidden feature dimension of 256.

## 5 EXPERIMENTS

We evaluate our method in simulation and the real world. In the simulation, we simulate a quadruped robot in a set of challenging and diverse environments. In the real world, we conduct experiments in indoor scenarios with obstacles and in-the-wild with complex terrain and novel obstacles.

### 5.1 ENVIRONMENTS IN SIMULATION

We design 6 simulated environments with varying terrain, obstacles to avoid, and spheres to collect for reward bonuses. Spheres are added to see whether agents are able to distinguish objects and their associated functions based on their appearance. All obstacles and spheres are randomly initialized and remain static throughout the episode unless stated otherwise. Specifically, our environments include: **Wide Obstacle** (Wide Obs.): wide cuboid obstacles on flat terrain, *without* spheres; **Thin Obstacle** (Thin Obs.): numerous thin cuboid obstacles on flat terrain, *without* spheres; **Wide Obstacle & Sphere** (Wide Obs.& Sph.): wide cuboid obstacles on flat terrain, including spheres that give a reward bonus when collected; **Thin Obstacle & Sphere** (Thin Obs.& Sph.): numerous thin cuboid obstacles and spheres on a flat terrain; **Moving Obstacle**: similar to the *Thin Obs.* environment, but obstacles are now dynamically moving in random directions updated at a low frequency. **Mountain**: a rugged mountain range with a goal on the top of the mountain. We show 4 environments above in Figure 1, omitting Wide Obs. and Thin Obs. for simplicity. We provide further details on the observation and action space, specific reward function, and relevant hyper-parameters in Appendix A.

**Reward Function.** For all environments, we adopt the same reward function containing the following terms: (i) *Forward reward* incentivizing the robot to move forward along a task-specific direction, i.e. towards the goal position in the Mountain environment (visualized as the red sphere in Figure 1), or

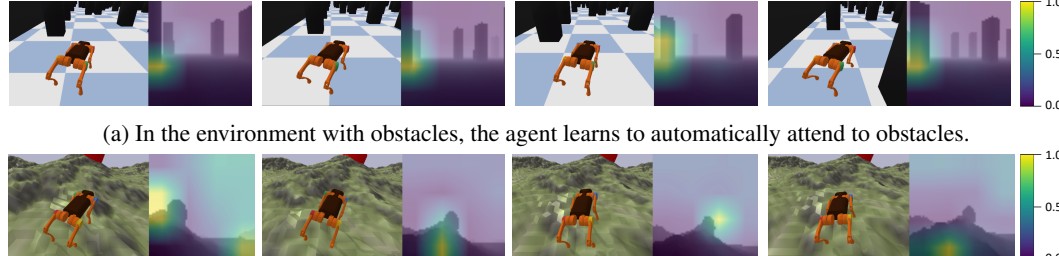

(a) In the environment with obstacles, the agent learns to automatically attend to obstacles.

(b) On challenging terrain, the agent attends to the goal destination and the local terrain in an alternative manner.

Figure 3: **Self-attention from our shared Transformer module.** We visualize the self-attention between the proprioceptive token and all visual tokens in the last layer of our Transformer model. We plot the attention weight over raw visual input where warmer color represents larger attention weight.

the move along the axis in all other environments (i.e. moving forward); (ii) *Sphere reward* for each sphere collected; (iii) *Alive reward* encouraging the agent to avoid unsafe situations, e.g. falling; and (iv) *Energy usage penalty* encouraging the agent to use motor torque of small magnitude.

## 5.2 BASELINE AND EXPERIMENTAL SETTING

To demonstrate the importance of visual information for locomotion in complex environments, as well as the effectiveness of our Transformer model, we compare our method to: *State-Only* baseline that only uses proprioceptive states; *Depth-Only* baseline that only uses visual observations; *State-Depth-Concat* that uses both proprioceptive states and vision, but without our proposed Transformer. The State-Depth-Concat baseline uses a linear projection to project visual features into a feature vector that has the same dimensions as the proprioceptive features. The State-Depth-Concat baseline then concatenates both features and feeds it into the value and policy networks. We also introduce a *Hierarchical Reinforcement Learning (HRL)* baseline as described in Jain et al. (2020), but without the use of the trajectory generator for a fair comparison (We follow Jain et al. (2020) faithfully and our results indicate that it works as expected). We train all agents using PPO (Schulman et al., 2017) and share the same proprioceptive and visual encoder for the value and policy network.

**Evaluation Metric and Training Samples.** We evaluate policies by their mean episode return, and two domain-specific evaluation metrics: (i) the distance (in meters) an agent moved along its target direction; and (ii) the number of collisions with obstacles per episode (with length of 1k steps). The collision is examined at every time step. , and we only compute the collision when the robot pass by at least one obstacle. We train all methods for 15M samples with 5 different random seeds and report the mean and standard deviation of the final policies.

## 5.3 ATTENTION MAPS

To gain insight into how our Transformer model leverages spatial information and recognizes dominant visual regions for decision-making at different time steps, we visualize the attention map of our policy on the simulated environment in Figure 3. Specifically, we compute the attention weight $W_{i,j}$ between the proprioceptive token and all other visual tokens and visualize the attention weights on the corresponding visual region of each token. In the top row, we observe that the agent pays most attention to nearby obstacles in the front, i.e. objects that the agent needs to avoid to move forward. The attention also evolves when new obstacles appear or get closer. In the Mountain environment (bottom row), the agent attends alternatively to two different types of regions: the close terrain immediately influencing the locomotion of the robot, and regions corresponding to the task-specific direction towards the target. The robot first attends to the terrain in front to step on the ground (1st & 3rd frame), once the agent is in a relatively stable state, it attends to the goal far away to perform longer-term planning (2nd & 4th frame). The regions attended by the agent are highly task-related and this indicates that our model learns to recognize important visual regions for decision-making.

## 5.4 NAVIGATION ON FLAT TERRAIN WITH OBSTACLES

**Static Obstacles without Spheres.** We train all methods on navigation tasks with obstacles and flat terrain to evaluate the effectiveness of modal fusion and stability of locomotion. Results are shown in Figure 4 (a). Our method, the HRL baseline, and the State-Depth-Concat baseline significantly outperform the State-Only baseline in both the *Thin Obstacle* and *Wide Obstacle* environment, demonstrating a clear benefit of vision for locomotion in complex environments. Interestingly, when

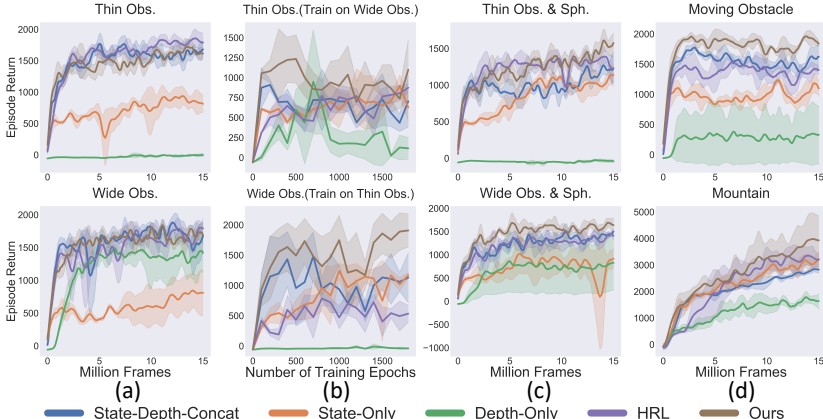

Figure 4: **Training and evaluation curves** on simulated environments (Concrete lines and shaded areas shows the mean and the std over 5 seeds, respectively). For environment without sphere (in (a)), our method achieve comparable training performance but much better evaluation performance on unseen environments (in (b)). For more challenging environment (in (c) and (d)) our method achieve better performance and sample efficiency.

Table 1: **Generalization.** We evaluate the generalization ability of all methods by evaluating on unseen environments. Our method significantly outperform baselines on both metrics (longer distance & less collision).

| | Distance Moved (m) $\uparrow$ | | Collision Happened $\downarrow$ | |
|---|---|---|---|---|
| | Thin Obs.(Train on Wide Obs.) | Wide Obs.(Train on Thin Obs.) | Thin Obs.(Train on Wide Obs.) | Wide Obs.(Train on Thin Obs.) |
| State-Only | $3.6_{\pm1.3}$ | $5.9_{\pm0.9}$ | $456.3_{\pm262.2}$ | $545.1_{\pm57.7}$ |
| Depth-Only | $1.1_{\pm1.1}$ | $0.1_{\pm0.0}$ | - | - |
| State-Depth-Concat | $5.6_{\pm2.1}$ | $7.1_{\pm2.0}$ | $406.8_{\pm89.5}$ | $331.1_{\pm192.8}$ |
| HRL | $5.8_{\pm2.2}$ | $11.5_{\pm1.8}$ | $527.9_{\pm94.6}$ | $238.8_{\pm59.5}$ |
| Ours | $\mathbf{8.2}_{\pm\mathbf{2.5}}$ | $\mathbf{14.2}_{\pm\mathbf{2.8}}$ | $\mathbf{310.4}_{\pm\mathbf{131.3}}$ | $\mathbf{82.2}_{\pm\mathbf{103.8}}$ |

the environment appearance is relatively simple (e.g., the *Wide Obstacle* environment), the Depth-Only baseline can learn a reasonable policy without using proprioceptive states. We surmise that the agent can infer part of the proprioceptive state from visual observations for policy learning. This phenomenon suggests that modeling the correlation between different modalities and better fusion techniques are essential for a good policy. We also observe that the simpler State-Depth-Concat baseline performs as well as our Transformer-based model in these environments. We conjecture that this is because differentiating obstacles from flat terrain is not a perceptually complex task, and a simple concatenation, therefore proves sufficient for policy learning.

We further evaluate the generalization ability of methods by transferring methods trained with thin obstacles to environments with wide obstacles, and vice versa. Figure 4 (b) shows generalization measured by episode return, and Table 1 shows average the quantitative evaluation results. While the State-Depth-Concat baseline is sufficient for training, we find that our Transformer-based method improves episode return in transfer by as much as **69**% and **56**% in the *Wide* and *Thin* obstacle environments, respectively, over the State-Depth-Concat baseline. Compared with the HRL baseline, the improvements of our method are **257.6**% and **118.2**%, respectively. We observe that our method moves significantly farther on average, and reduces the number of collisions by **290.5**%, **402**% and **663**% over the HRL baseline, the State-Depth-Concat and State-Only baselines when trained on thin obstacles and evaluated on wide obstacles. The Depth-Only baseline fails to generalize across environments and no collision occurs as the robot moves too little to even collide with obstacles. Interestingly, we observe that the generalization ability of the State-Depth-Concat *decreases* as training progresses, whereas for our method it either plateaus or *increases* over time. This indicates that our method is more effective at capturing essential information in the visual and proprioceptive information during training, and is less prone to overfit to training environments.

**Static Obstacles with Spheres.** We now consider a perceptually more challenging setting with the addition of spheres in the environment; results are shown in Figure 4 (c). We observe that with additional spheres, the sample efficiency of all methods decreases. While spheres with positive reward provide the possibility for higher episode return, spheres increase complexity in two ways: (i) spheres may lure agents into areas where it is prone to get stuck; and (ii) although spheres do not block the

Table 2: **Evaluation on environments with spheres.** We evaluate the final policy of all methods. Our method achieved the best performance on almost all environment for all metrics.

| | Distance Moved (m) ↑ | | Sphere Reward ↑ | | Collision Happened ↓ | |
| | Thin Obs. & Sph. | Wide Obs. & Sph. | Thin Obs. & Sph. | Wide Obs. & Sph. | Thin Obs. & Sph. | Wide Obs. & Sph. |
|---|---|---|---|---|---|---|
| State-Only | $5.6_{\pm 1.6}$ | $7.4_{\pm 2.8}$ | $80.0_{\pm 43.2}$ | $80.0_{\pm 32.7}$ | $450.2_{\pm 59.7}$ | $556.5_{\pm 173.1}$ |
| Depth-Only | $0.0_{\pm 0.1}$ | $5.2_{\pm 3.9}$ | $0.0_{\pm 0.0}$ | $33.3_{\pm 47.1}$ | - | - |
| State-Depth-Concat | $13.1_{\pm 2.3}$ | $11.4_{\pm 3.3}$ | $206.0_{\pm 41.1}$ | $193.3_{\pm 24.9}$ | $\mathbf{229.2_{\pm 65.3}}$ | $87.2_{\pm 40.7}$ |
| HRL | $10.8_{\pm 0.8}$ | $11.3_{\pm 2.9}$ | $166.7_{\pm 54.4}$ | $288.9_{\pm 154.8}$ | $256.8_{\pm 87.4}$ | $423.3_{\pm 170.0}$ |
| Ours | $\mathbf{15.2_{\pm 1.8}}$ | $\mathbf{14.5_{\pm 0.7}}$ | $\mathbf{233.3_{\pm 47.1}}$ | $\mathbf{220.0_{\pm 33.2}}$ | $256.2_{\pm 70.0}$ | $\mathbf{54.6_{\pm 20.8}}$ |

agent physically, they may occlude the agent's vision and can be visually difficult to distinguish from obstacles in a depth map. We observe that with increased environmental complexity, our method consistently outperforms both the HRL baseline and the State-Depth-Concat baseline in the final performance and sample efficiency. We report the average distance moved, number of collisions, and the reward obtained from collecting spheres, in Table 2. Our method obtains a comparable sphere reward but a longer moved distance, which indicates that our LocoTransformer method is more capable of modeling complex environments using spatial and cross-modal attention.

**Moving Obstacles.** When the positions of obstacles are fixed within an episode, the agent may learn to only attend to the closest obstacle, instead of learning to plan long-term. To evaluate the ability of long-term planning, we conduct a comparison in an environment with moving obstacles to simulate real-world scenarios with moving objects like navigating in the human crowd. The top row of Figure 4 (d) and Table 3 shows that the State-Only baseline and the Depth-Only baseline both perform poorly, and

Table 3: **Evaluation results** on the Moving Obstacle Environment.

| Method | Distance Moved (m) ↑ | Collision Happened ↓ |
|---|---|---|
| State-Only | $6.0_{\pm 1.3}$ | $129.4_{\pm 25.4}$ |
| Depth-Only | $1.1_{\pm 1.1}$ | - |
| State-Depth-Concat | $\mathbf{16.3_{\pm 1.7}}$ | $88.4_{\pm 34.0}$ |
| HRL | $7.1_{\pm 2.6}$ | $75.8_{\pm 11.0}$ |
| Ours | $11.3_{\pm 2.9}$ | $\mathbf{67.9_{\pm 18.1}}$ |

the HRL baseline performs worse than the State-Depth-Concat baseline. These results indicate that the State-Only baseline lacks planning skills, which can be provided by visual observations, and the hierarchical policy can not fuse the information from different modalities effectively when the environment is sufficiently complex. While the State-Depth-Concat baseline performs better in terms of distance, it collides more frequently than our method. This indicates that the baseline fails to recognize the moving obstacles, while our method predicts the movement of obstacles and takes a detour to avoid potential collisions. In this case, the conservative policy obtained by our method achieved better performance in terms of episode return though it did not move farther. We deduce that with only a compact visual feature vector, it is very hard for the State-Depth-Concat baseline to keep track of the movement of obstacles in the environment. On the other hand, it is easier to learn and predict the movement of multiple obstacles with our method since the Transformer provides an attention mechanism to model the visual region relations.

**Ablations.** We evaluate the importance of two components of our Transformer model on the Thin Obs. & Sph. environment: (1) the number of Transformer encoder layers; and (2) the number of visual tokens ($N^2$ visual tokens). Results are shown in Table 4. From Table 4b, we observe that the performance of our model is relatively insensitive to the number of

Table 4: **Ablation study on Thin Obs. & Sph.**: We perform ablations on Thin Obs. & Sph. environment and adopt the best setting ($N = 4, L = 2$) for all environments, which includes 16 visual tokens and 2 Transformer encoder layers.

(a) On Number of Visual Tokens

| Method | Episode Return ↑ |
|---|---|
| Ours (N=1) | $1204.8_{\pm 243.6}$ |
| Ours (N=2) | $1418.1_{\pm 167.8}$ |
| Ours (N=4) | $\mathbf{1551.5_{\pm 120.4}}$ |

(b) On Number of Layers

| Method | Episode Return ↑ |
|---|---|
| Ours (L=1) | $1509.7_{\pm 244.8}$ |
| Ours (L=2) | $\mathbf{1551.5_{\pm 120.4}}$ |
| Ours (L=3) | $1423.5_{\pm 100.7}$ |

Transformer encoder layers. For ablation on the number of visual tokens, we change the kernel size and the stride of the last convolutional layer in our visual encoder to get visual features with different shapes and different numbers of visual tokens. From Table 4a, we can see that the performance of our method is positively correlated with the number of the visual tokens. With a fixed size of the visual feature map, a higher number of tokens directly results in a smaller receptive field for each visual token. Because our method performs spatial cross- modality attention across all tokens, our model benefits from richer low-level visual information. This indicates a potential for our model to work with high-resolution visual input and in more complicated environments and complex tasks.

## 5.5 Navigation on Simulated Uneven Terrain

We also evaluate all methods on uneven, mountainous terrain. The bottom row of Figure 4 (d) and Table 5 shows training curves and the mean distance moved for each method, and our method improves over all baselines by a large margin in episode return. Despite having access to depth images, the State-Depth-Concat baseline does not show any improvement over the State-Only baseline in episode return. We conjecture that naively projecting spatial-visual feature into a vector and fusing multi-modality information with a simple concatenation can easily lose the spatial structure of visual information. Although

Table 5: **Evaluation Result** on the Mountain environment.

| Method | 3D Distance Moved (m) ↑ |
|---|---|
| State-Only | $3.7_{\pm 1.6}$ |
| Depth-Only | $3.0_{\pm 0.5}$ |
| State-Depth-Concat | $4.7_{\pm 0.8}$ |
| HRL | $6.3_{\pm 0.3}$ |
| Ours | $\mathbf{6.8}_{\pm 1.1}$ |

the HRL baseline moves farther among baselines, it does not obtain a higher episode return, indicating the HRL baseline is not able to utilize the visual guidance towards the target. Our Transformer-based method better captures spatial information such as both global and local characteristics of the terrain and more successfully fuses spatial and proprioceptive information than a simple concatenation.

## 5.6 Real-world Experiments

To validate our method in different real-world scenes beyond the simulation, we conduct real-world experiments in both indoor scenarios with obstacles (referred as **Indoor & Obs.**) and in-the-wild forest with complex terrain and trees (referred to as **Forest**)) as shown in Figure 5. As the HRL baseline is found to not generalize well to unseen environments as shown in Figure 4 (b) and Table 1, we only deploy policies learned in simulation using our LocoTransformer and the State-Depth-Concat baseline on a Unitree A1 Robot (Unitree, 2018). The policies are trained with the Thin Obstacle environment randomized with uneven terrains. All the real-world deployment experiments are repeated **15** times across different seeds. Details about robot setup are provided in Appendix A. Since it is challenging to measure the exact duration of collision with obstacles in the real world, we instead report the number of times that robot collides with obstacles (**Collision Count**) as a measure of performance.

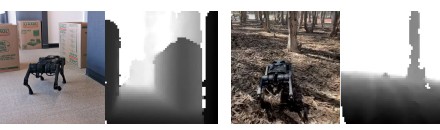

(a) Indoor & Obs.   (b) Forest

Figure 5: **Real World Samples** We evaluate our method in real-world scenarios with different obstacles on complex terrain.

Figure 6: **Experiment results in the real-world**: We perform real-world experiment on Indoor & Obs. and Forest environments.

| Method | Distance Moved (m) ↑ | Collision Times ↓ |
|---|---|---|
| State-Depth-Concat | $5.0_{\pm 2.6}$ | $0.4_{\pm 0.5}$ |
| Ours | $\mathbf{9.6}_{\pm 2.2}$ | $\mathbf{0.3}_{\pm 0.5}$ |

(a) Indoor & Obs.

| Method | Distance Moved (m) ↑ | Collision Count ↓ |
|---|---|---|
| State-Depth-Concat | $5.1_{\pm 0.9}$ | $0.3_{\pm 0.5}$ |
| Ours | $\mathbf{9.6}_{\pm 2.0}$ | $\mathbf{0.0}_{\pm 0.0}$ |

(b) Forest

As shown in Table 6, our method outperforms the baseline by a large margin in both scenarios. In the Indoor & Obs environment, our method moves **92%** farther than the baseline and collides less. When facing complex terrain and unseen obstacles in the Forest environment, our method greatly improves over the baseline; our policy moved approximately **90%** farther without colliding into any obstacles, while the baseline frequently collides into trees and gets stuck in potholes. We generally observe that our method is more robust than the baseline when deployed in the real world, indicating that our method better captures the object structure from visual observations, rather than overfitting the appearance of objects during training.

## 6 Conclusion

We propose to incorporate the proprioceptive and visual information with the proposed LocoTransformer model for locomotion control. By borrowing the visual inputs, we show that the robot can plan to walk through different sizes of obstacles and even moving obstacles. The visual inputs also helps the locomotion in challenging terrain such as mountain. Beyond the training environment, we also show that our method with the cross-modality Transformer achieves better generalization results when testing on unseen environments and in the real world. This shows our Transformer model provides an effective fusion mechanism between proprioceptive and visual information and new possibilities on reinforcement learning with information from multi-modality.

## 7 REPRODUCIBILITY STATEMENT

To ensure the reproducibility of our work, we provide the following illustrations in our paper and appendix:

- **Environment**: We provide the detailed description of the environment in Section 5.1, as well as the specific observation space, action space and reward function in Appendix A.2.
- **Implementation Details**: We provide all implementation details and related hyperparameters for both our methods and baselines in Section 4.2 and Appendix B.
- **Real Robot Setup**: We provide all relavant details about setting up the real robot and conduct real-world experiment in Appendix A.3.

We have released the code, environment and videos on our project page: `https://rchalyang.github.io/LocoTransformer/`. We believe the open source of our code and environment will be an important contribution to the community.

**Acknowledgement**: This work was supported, in part, by gifts from Meta, Qualcomm, and TuSimple.

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

## A    Detailed Experiment Setup

### A.1    Details on Proprioception and Action

Our Unitree A1 robot has 12 Degrees of Freedom (DoF), and we use position control to set actions for the robot. Specifically, the proprioceptive input contains the following components:

- Joint angle: a 12-dimensional vector records the angle of each joint.

- IMU information: a 4-dimensional vector records orientations and angular velocities.

- Base displacement: a 3-dimensional vector records the absolute base position of robot.

- Last action: a 12-dimensional vector records the angle change in the last step.

The full proprioceptive vector consists of all these vectors over the last three steps to retain historical state information. The action is also a 12-dimensional vector that controls the change of all the joint angles. We use 0.5 as the upper bound of action for locomotion stability. We use all default settings of A1 robot in the official repository.

### A.2    Reward Definition

In all our experiments, we use the same reward function as follow:

$$R = \alpha_{\text{forward}} R_{\text{forward}} + \alpha_{\text{energy}} R_{\text{energy}} + \alpha_{\text{alive}} R_{\text{alive}} + K \cdot R_{\text{sphere}}, \tag{7}$$

where we set $\alpha_{\text{forward}} = 1, \alpha_{\text{energy}} = 0.005, \alpha_{\text{alive}} = 0.1$ for all tasks.

$R_{\text{forward}}$ stands for moving forward reward. In flat environments, it's defined by the moving speed of robot along the x-axis; in mountain environment, it's defined by that along the direction to the mountain top (red sphere in Figure 1 Mountain in paper).

$R_{\text{energy}}$ ensures the robot is using minimal energy, which has been shown to improve the naturalness of motion, similar to Yu et al. (2018). Specifically, we penalize the actions resulting motor torques with large euclidean norm.:

$$R_{\text{energy}} = -\|\tau\|^2, \quad \tau \text{ is the motor torques.}$$

$R_{\text{alive}}$ encourages the agent to live longer. It gives a positive reward of $1.0$ at each time step until termination. Dangerous behaviors like falling down and crashing into obstacles will call termination.

$R_{\text{sphere}}$ stands for sphere collection reward (whenever applicable) for each sphere collected, and $K$ is the number of spheres collected at the current time step.

### A.3    Real Robot Setup

We use the Unitree A1 Robot (Unitree, 2018), which has 18 links and 12 degrees of freedom (3 for each leg). We mount an Intel RealSense camera at the head of the robot to capture the depth map, and use the robot sensors to get the joint states and IMU for the proprioceptive input. All computations are running with on-board resources. We set the control frequency to be 25 Hz and set the action repeat to be 16, so that the PD controller converts the target position commands to motor torques at 400 Hz. We set the KP and KD of PD controller to be 40 and 0.6 respectively. The base displacement is removed from the observation for the real-world experiment. For the real-world experiment, we execute the policy on the real-robot for **20** seconds and measure the Euclidean distance between the start and end point of the trajectory for evaluating the performance.

## B    Hyperparameters

In this section, we detail the hyperparameters for each method used in our experiments.

| Parameters | Range |
|---|---|
| KP | [40, 90] |
| KD | [0.4, 0.8] |
| Inertia ($\times$ default value) | [0.5, 1.5] |
| Lateral Friction (Ns / m) | [0.5, 1.25] |
| Mass ($\times$ default value) | [0.8, 1.2] |
| Motor Friction (Nms / rad) | [0.0, 0.05] |
| Motor Strength ($\times$ default value) | [0.8, 1.2] |
| Sensor Latency (s) | [0, 0.04] |

Table 6: Variation of Environment and Robot Parameters.

## B.1 DOMAIN RANDOMIZATION

To narrow the reality gap, we leverage domain randomization during training phase. All methods conducted in real world experiments use a same group of randomization setting to be fair. Specifically, we set the range of parameters as follow:

During training, besides the domain randomization for the proprioceptive state, we perform domain randomization for visual input. Specifically, at each time-step, we randomly choose 3 to 30 values in $(64, 64)$ depth input and set the depth reading to the maximum reading. In this case, we simulate the noisy visual observation in the real-world.

## B.2 HYPERPARAMETERS SHARED BY ALL METHODS

Here we give the details of all hyperparameters that are related to reinforcement learning and shared by all tested methods. "Horizon" denotes the episode length in both training and testing, and "Clip parameter" denotes the max norm of gradients in all trained networks.

| Hyperparameter | Value |
|---|---|
| Horizon | 1000 |
| Non-linearity | ReLU |
| Policy initialization | Standard Gaussian |
| # of samples per iteration | 8192 |
| Discount factor | .99 |
| Batch size | 256 |
| Optimization epochs | 3 |
| Clip parameter | 0.2 |
| Policy network learning rate | 1e-4 |
| Value network learning rate | 1e-4 |
| Optimizer | Adam |

## B.3 STATE-ONLY BASELINE

To keep the comparison fair, we use 4 fully-connected layers for state-only baseline to keep the network size large enough for higher learning capacity. We also try more layers but observe minor difference in performance.

| Hyperparameter | Value |
|---|---|
| Network size | 4 FC layers with 256 units |

## B.4 STATE-DEPTH-CONCAT BASELINE

Apart from the perception encoder, we keep all other parts same for State-Depth-Concat baseline and our LocoTransformer.

| Hyperparameter | Value |
|---|---|
| Proprioceptive encoder | 2 FC layers with 256 units |
| Projection Head | 2 FC layers with 256 units |

## B.5    OURS - LOCOTRANSFORMER

| Hyperparameter | Value |
|---|---|
| Token dimension | 128 |
| Proprioceptive encoder | 2 FC layers with 256 units |
| Projection Head | 2 FC layers with 256 units |
| # of transoformer encoder layers | 2 |

## C    MORE ATTENTION VISUALIZATION RESULTS

We offer more visualization of attention maps here to show that LocoTransformer consistently attends to reasonable parts of environment during an episode. In the *Thin Obstacle*, robot will be aware of the new appearing obstacles and give emergent reaction to escape the threat. For example, in the last row, the robot attends to the closed obstacle and then turns left. Suddenly it attends to the wall, so it reorientates its body to avoid risky. In the *Mountain*, robot not only pays attention to the final goal position, but also attend to rugged terrains that are extremely hard to step on. This also shows the planning ability according to different visual regions learned by Transformer architecture. In the second case, the robot first attends to goal position to ensure the forward direction, but the uneven rocks also draw its attention.

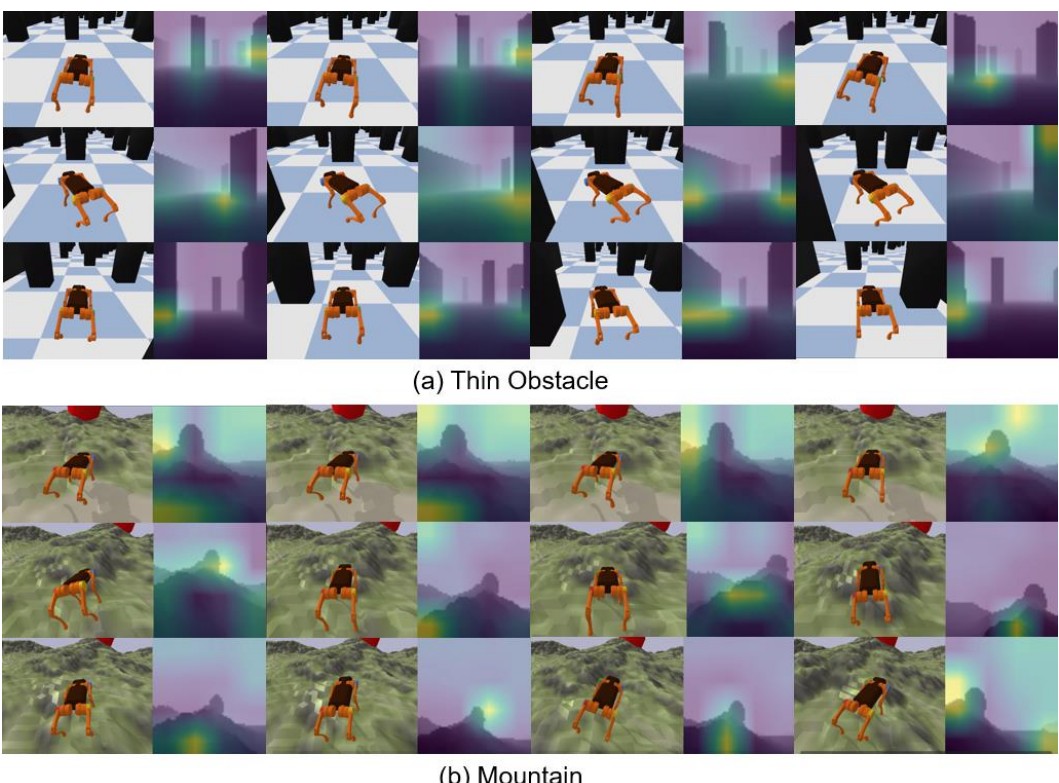

(a) Thin Obstacle

(b) Mountain

Figure 7: **Additional Attention Visualization** We visualize more attention map visualization for better understanding of how our LocoTransformer works. Each row shows a sequence of attention map to present how the attention of agent evolves.

## D    COMPARISON WITH MODEL PREDICTIVE CONTROL

To understand the advantages of learning-based locomotion, we offer a detailed comparison with the classical quadrupedal robotic control pipeline, which is commonly used in both classicalal approaches and recent visual locomotion learning works. With this comparison, we can also help the community understand the main difference between learning-based academic works and industrial solutions, like the famous Boston Dynamics Spot.

### D.1 VISION-GUIDED WHOLE-BODY CONTROLLER

We mainly follow Carlo et al. (2018) and the code for (Peng et al., 2020) to reproduce the vision-guided whole-body controller for A1. Our high-level vision controller is trained with RL perceiviing the visual information same as other approaches and outputing the target linear and angular velocity for low-level controller. Our low-level motion controller provide the actual motor commands to the robot according to current target velocity.

**High-level Visual Policy.**  High-level visual policy outputs the target linear velocity and angular velocity given the stacked depth maps and (optionally) CoM velocity and IMU information. If only given the depth maps, we remove the state encoder part in the original models and keep the Transformer and CNN encoders; if given both the depth maps and the body state (CoM velocity and IMU information), we just keep the two architectures same as the main paper. We train the high-level controller with PPO to provide fair and consistent comparison with our method. The control frequency of the high-level controller is 20Hz, with the action repeat set as 10 to guide the low-level controller. We use the Unicycle Model to control the CoM velocity, i.e., specifying the absolute linear velocity and angular (rotating) velocity. In our experiments, the target linear velocity is clipped to $\pm 0.4$ and the target angular velocity is clipped to $\pm 0.3$.

**Low-level Controller.**  The low-level controller uses position control for swing actions and torque control for stance actions. Specifically, we use a finite state machine (FSM) based gait scheduler to decide when to swing each leg and how long stance each leg needs in a complete control cycle. The swing action is determined by a fixed foot clearance height and controlled by a PD-controller with the target foot position. The stance force (torques of each joint in a leg) is computed by model predictive control to track the desired CoM velocity. The whole-body controller outputs a new command in 200HZ and use another action repeat 5 to control the body.

**Training.**  We use PPO to train the control policy, similar to other methods in this paper. Here we only demonstrate some main modifications that might influence reproducing the experiments: 1) we remove the energy reward term, since the low-level motion control is not learnable; 2) Because of the change of the control frequency (all other methods in our paper directly provide low-level command which requires a higher control frequency), we tune the following hyperparameters:

| Hyperparameter | Value |
|---|---|
| Horizon | 500 |
| # of samples per iteration | 4096 |
| $\alpha_{\text{forward}}$ | 0.5 |

We remove domain randomization terms which are only related to motion robustness.

### D.2 RESULTS IN SIMULATION

We perform the training in simulation and answer the following two questions:

- *Can our Transformer-based architecture still outperform CNN-based one while combining with classical controller?*
- *Is our RL-based framework better than visual policy + classical controller?*

Our results show the advantage of our network design and that of our End-to-end RL pipeline.

**Comparison between Network Architectures.**  We train the vision-guided whole body controller in two settings. The two settings are: (i) Stacked depth maps; (ii) Stacked depth maps with CoM velocity and IMU sensor input. The training curves are in Figure 9. Due to the difference of the reward settings and episode length, it's meaningless and unfair to compare the curves with that of RL-based methods. The training results show that for multi-modal setting, our LocoTransformer still outperforms the baseline. For vision-only setting, we observe minor difference between transformer and CNN, and speculate that attention mechanism indeed improves the modal fusion rather than image representation learning.

Note that it costs several times of samples to train the controller. We deduce the phenomenon is due to low-level controller is not agile enough to the varying high-level command to gain explicit information for RL optimization.

For customized traditional controllers, such as the MPC controller used by MIT cheetah 3 (Carlo et al., 2018), the inference speed is at around 120Hz on the Unitree A1 robot. To overcome the limitation of computation, an external laptop is used in (Sun et al., 2021; Yang et al., 2021). In contrast, our end-2-end learned policy is able to utilize the on-board GPU to perform real-time inference.

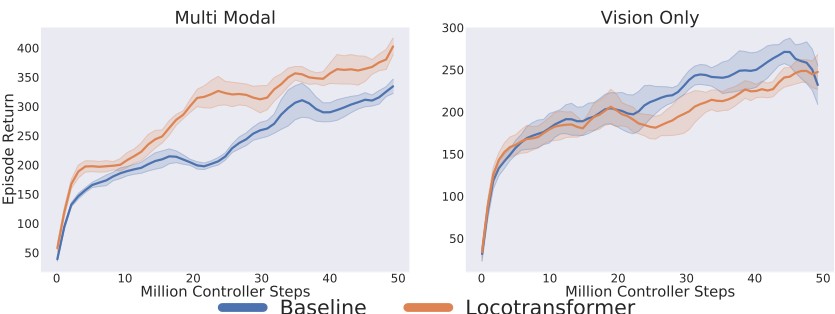

Figure 8: **Training curves of vision-guided whole body controller.** For multi-modal input setting, Loco-Transformer still outperforms the baseline.

## D.3 RESULTS IN REAL WORLD

To further illustrate the advantage of end-to-end RL for locomotion, we deploy the vision-guided whole-body controller using only visual observation and baseline architecture in the real world. The vision-guided whole-body controller controller generates natural and stable gaits, including larger amplitude of feet swing, lower frequency of stepping, and consistent body height. However, when facing obstacles like trees, the vision-guided whole-body controller cannot plan well with the MPC controller for avoidance. It tries to avoid the tree, but since the motion is not very flexible and the high-level transitions are not smooth, it collides with the side of the tree, and bounces away to come across the obstacle. While the agent is still moving forward, having more collisions makes the robot unsafe. On the other hand, when trained end-to-end with both vision and legged motion, our approach can flexibly change from a moving forward motion to a turning motion and adjust the speed at the same time. We generally observe much more diverse motions emerge and our policy is able to transition smoothly between these motions according to vision. For in-door environments where the obstacles are larger, there are less chances for the vision-guided whole-body controller to come across the obstacle when the robot is about to collide into it.

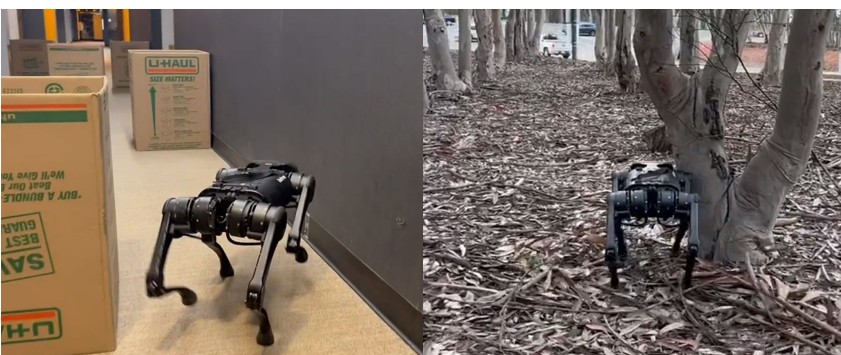

Figure 9: **Some failure cases of vision-guided whole-body controller due to limitation of agility.** The left figure shows that when walking through a narrow path, the robot can not quickly turn around and collide into the wall. The right figure denotes that it may collide to the tree in the wild, due to the shape of the obstacles are out of the training distribution and the robot can't adjust quickly enough to avoid.

## D.4 Quantitative Comparison in Real World

We also provide quantitative comparisons between our method and the vision-guided whole-body controller. The vision-guided whole-body controller create more collisions even though this helps walking longer distances.

Figure 10: **Experiment results in the real-world**: We perform real-world experiment on Indoor & Obs. and Forest environments to compare our method and the vision-guided whole-body controller

(a) Indoor & Obs.

| Method | Distance Moved (m) ↑ | Collision Times ↓ |
|---|---|---|
| Vision-Guided Whole-Body Controller | $7.5_{\pm 1.5}$ | $3.4_{\pm 0.8}$ |
| Ours | $\mathbf{9.6_{\pm 2.2}}$ | $\mathbf{0.3_{\pm 0.5}}$ |

(b) Forest

| Method | Distance Moved (m) ↑ | Collision Count ↓ |
|---|---|---|
| Vision-Guided Whole-Body Controller | $\mathbf{11.6_{\pm 3.5}}$ | $1.9_{\pm 0.8}$ |
| Ours | $9.6_{\pm 2.0}$ | $\mathbf{0.0_{\pm 0.0}}$ |

For in-door environments where the obstacles are larger, there is less chance for the vision-guided whole-body controller to avoid the obstacle when the robot is about to collide into it. Thus the vision-guided whole-body controller performs worse in both collision times and distance moved compared to our approach.

