# OpenReview forum: "Learning Vision-Guided Quadrupedal Locomotion End-to-End with Cross-Modal Transformers"
_ICLR.cc/2022/Conference — ICLR 2022 Spotlight_

### Official Review · Reviewer_Dcnh · 2021-10-26

**Correctness:** 4
**Technical Novelty And Significance:** 3
**Empirical Novelty And Significance:** 3
**Recommendation:** 8
**Confidence:** 5

**Main Review:**

This paper is very clear in its exposition, providing a detailed diagram of the method, clearly labeled inputs and outputs, and relevant implementation details. The experiments appear sound, with a number of baselines and ablations provided. I want to emphasize the value of real-world experiments in this space, as the 'sim-to-real' gap can be significant and invalidate otherwise good looking results. Scientifically, the paper proposes an architecture that is novel and can serve as a broader proof point that vision-based locomotion can be competitive and robust when trained with a sufficiently expressive model.

Strengths:
+ The paper tackles an important research problem: vision-guided locomotion and has made significant progress along this direction.
+ Novel network architectures (such as Transformers) are under-explored in the legged robots community. This paper demonstrates that incorporating such architecture indeed makes a difference in performance.
+ The proposed method is validated on a real robot. The evaluations are comprehensive and the conclusions are convincing.

Weakness:
- The technical novelty is lean. Neither of the key components of the paper are novel: RL for locomotion and Transformers. Although this can be considered as a weakness, it is not a deal breaker given that the combination of these two and the application to legged robots are novel and potentially influential.
- The results are mostly obstacle avoidance on the flat ground, where legs are not essential. Most of the experiments can be done with a wheeled robot. To show the true value of this paper, more challenging terrain needs to be considered and tested on, such as stairs, stepstones, tall grasses, rocks, etc. In these terrains, both vision and legs are critical. From the accompanying video, it is a bit disappointing that the robot only learns steering for obstacle avoidance, but does not learn foot clearance or foothold location on different types of terrains. The robot always drags the foot, even on pebbles (5:25s in the video) where higher foot clearance is clearly a preferred choice. Since this paper trains end-to-end, I would expect that these behaviors would emerge automatically if trained in relevant environments. Showing these behaviors (foot clearance, foothold location, change of gait pattern) in addition to obstacle avoidance, would significantly strengthen the paper.

Additional questions:
1) In addition to domain randomization, does the paper apply other techniques for sim-to-real transfer? For example, I would imagine that there will be a large sim-to-real gap in vision. The depth images from Intel realsense can be noisy and with holes, especially in outdoor environments. Do these sim-to-real gaps in vision cause any problems when deploying the policy on the robot?
2) How much tuning is needed to learn natural (deployable) locomotion gaits? In the video, the learned locomotion gait is quite reasonable and deployable on the robot. The paper explicitly mentioned that it did not use trajectory generators. Does it purely rely on reward shaping? If so, how much tuning is needed? And what are the most important terms in the reward function that encourage the emergence of natural gaits?


**Summary Of The Paper:**

This paper proposes an approach to legged locomotion which leverages a Transformer-based model and is trained via end-to-end reinforcement learning. It provides extensive experimental evaluation of the approach in terms of performance and safety metrics, both in simulation and using real-world experiments. The code is expected to be open-sourced.

**Summary Of The Review:**

Good paper on relevance and experimental evidence, on a topic that is very much of interest to the robot learning community today.
Novelty limited due to combination of known techniques.
EDIT: bumped confidence to a 5 based on comments and rebuttal. This is a solid contribution.

---

> ### Author Response · Authors · 2021-11-21
> **For Reviewer Dcnh**
>
> Dear reviewer, thank you for all the detailed comments and suggestions. To address your concern, we would like to provide response to your comments here:
>
> **Q:** *“The technical novelty is lean... Although this can be considered as a weakness, it is not a deal-breaker given that the combination of these two and the application to legged robots are novel and potentially influential.”*
>
> **A:** Thank you for acknowledging our contributions on applying transformers for vision-guided locomotion. We believe how to integrate the multi-modal information (proprioceptive states, depth sensor observations) is important for locomotion control, going beyond blind robots.
>
> **Q:** *“The results are mostly obstacle avoidance on the flat ground, where legs are not essential...the robot only learns steering for obstacle avoidance, but does not learn foot clearance or foothold location on different types of terrains.”*
>
> **A:** Thank you for looking into our videos and pointing out the problem.
>
> First, we would like to emphasize that the forest environment we perform experiments on is not that trivial. As shown in our new video (https://drive.google.com/file/d/1yZ6pB3Cjdg7w_AAbkE6dR6rT8OCNjDc4/view?usp=sharing ), an MPC planner gets stuck and falls on the sticks but our method manages to maneuver over. We believe learning robust and flexible locomotion skills are essential for this environment.
>
> Second, we agree with the reviewer that our current robot is not always operating with foot clearance conditioning on the given terrain. The main reason is that some of the changes in the environment might not be so apparent when we observe from the depth image taken from the front head, facing to the forward direction. One possible solution is to add more sensory inputs, for example, one more camera facing down to the ground so that more detailed information about the terrain can be captured. Adding more sensory inputs can be a future work direction for our paper. We believe our current RL approach with multi-modal Transformer has its merits and shows great potential in this direction of research.
>
> **Q:** *“In addition to domain randomization, does the paper apply other techniques for sim-to-real transfer? For example, I would imagine that there will be a large sim-to-real gap in vision. The depth images from Intel realsense can be noisy and with holes, especially in outdoor environments. Do these sim-to-real gaps in vision cause any problems when deploying the policy on the robot?”*
>
> **A:** Yes, there is indeed Sim2Real gap on depth sensors. During training, besides the domain randomization for the proprioceptive state, we perform domain randomization for visual input. Specifically, we add random noise and holes in the visual observation to simulate the noisy visual observation in the real world. We provide the specific details in Appendix B in the revised paper.
>
> In addition, we find our Transformer model enhances the robustness of the policy. Figure 6 in the paper shows there is a large gap between policy with / without using the Transformer. The possible reasons are: (i) Information propagation between visual tokens can enhance the understanding of the depth image and compensate the missing regions; (ii) The mutual reasoning between the state token and the visual tokens help the policy on focusing the important regions (as shown in Figure 3), which at the same time reduces the effects of noise in most regions.
>
> **Q:** *“How much tuning is needed to learn natural (deployable) locomotion gaits? Does it purely rely on reward shaping? If so, how much tuning is needed? And what are the most important terms in the reward function that encourage the emergence of natural gaits?”*
>
> **A:** To learn a natural and deployable locomotion policy, we perform reward shaping and add multiple constraints in the environment. For reward function (see Appendix A.2), ours is relatively simple compared to most locomotion works using RL. Among these rewards, the most important term for gait emerging is the energy reward, which encourages the robot to perform locomotion while minimizing the energy consumption. This can lead to natural and deployable gait for the following reasons: (i) The energy reward prevents high joint speed, which could potentially damage the motor of the robot; (ii) It is inspired by the fact that the biological system in the real-world tends to use energy-efficient gait to fulfill the task.
>
> In addition, we have the following constraint for the environment during training to enable the robot to learn a natural locomotion gait: (i) We terminate the episode when the height of the center of the robot is higher or lower than the specified threshold; (ii) We clip the action when it achieves a high value.

---

> > ### Comment · Reviewer_Dcnh · 2021-11-21
> > **Thank you for the response.**
> >
> > Nice work. Increased my assessment confidence score.

---

### Official Review · Reviewer_GRoY · 2021-11-01

**Correctness:** 4
**Technical Novelty And Significance:** 3
**Empirical Novelty And Significance:** 3
**Recommendation:** 8
**Confidence:** 4

**Main Review:**

## Strengths and Weaknesses

### Things I liked about this paper
- **a powerful framework**: fusing visual and proprioceptive data for quadrupedal locomotion using transformer architectures is an interesting and also valuable approach that works well and sets an excellent opportunity for future work.
- **a well written paper**: overall the paper is well written and all sections are broadly very clearly described.
- **useful insight**: I like the provided key insight that proprioceptive states offer contact measurements for immediate reaction while visual sensory observations can help with longer-term planning.

### Things that can be improved
- **number of seeds is not great**: The adopted model-free approach is known to have very unstable learning process of the dynamics function which ideally requires 10 or more seeds to provide a solid results. Using only 5 seeds is not great. More details below.
- **prose is not perfect**: There are some minor details and clarifications that may help further improve clarity.

Using 5 random seeds for a model free approach is rather small as a number. Ideally, the evaluation should be done on 10 or more seeds. In fact, I suspect that some of the results such as the moving obstacles from Table 3, would change if the approach was evaluated on more runs. Nevertheless, the provided training curves seem to have fairly small variance as illustrated in Figure 4 which makes me more inclined to agree that 5 seeds are sufficient to report on accurate results. In addition, I would expect that the variation in the learnt dynamics to primarily affect the performance of the learnt agent on the physical quadruped system.  However, this does not seem to be the case in the reported results, which is great as long as all 5 seeds were used to extract those results. It is great that the paper considers 15 runs per seed but I wonder if the results were acquired through cherry picking best n seeds. This is a detail that is not currently mentioned in the paper but would certainly improve clarity if it did.

There are a few additional minor comments. Currently, the distance measurement reported in meters is mentioned only in the text and not in the tables. Stating this there too would make it much clearer. Similarly, what exactly does the collision happened represent. Are these total number of collisions over 1000 steps? 'the number of time steps where collision happens between the robot and obstacles
over the course of an episode' states the explanation seems a bit overly complicated. Why not just 'the number of collisions with obstacles per 1k step long episode' or something along those lines?

There is a typo in the contribution 'we the propose' should be 'we propose'. Another typo is '... whereas it for our method either plateaus' should be 'whereas for our method it either plateaus'

**Summary Of The Paper:**

This work proposes a novel architecture for quadrupedal locomotion that fuses proprioceptive and visual information with a transformer-based model to enable an agent to proactively maneuver environments with obstacles and uneven terrain by anticipating changes in the environment many steps ahead. The method is extensively evaluated in simulation and on a sim to real transfer tasks. The method is shown to both achieve higher reward  but also better capacity to generalise in the context of sim to real. Overall, the paper is well written and the provided evaluation is conducted fairly and well.

**Summary Of The Review:**

Overall this paper is written well and has a sound idea that is supported well by an extensive evaluation. There are some minor details that may further improve the quality of the paper but I see this as a strong submission which I can recommend for acceptance.

---

> ### Author Response · Authors · 2021-11-21
> **For Reviewer GRoY**
>
> Dear reviewer, thank you for all the detailed comments and suggestions. To address your concern, we would like to provide response to your comments here:
>
> **Q:** *“Number of seeds is not great: ... ideally requires 10 or more seeds to provide a solid result. Using only 5 seeds is not great…the moving obstacles from Table 3, would change if the approach was evaluated on more runs.”*
>
> **A:** We first emphasize that we **DID NOT** cherry pick the best seeds in all of our experiments. We use the same set of seeds for all methods. We re-run our LocoTransformer and the State-Depth-Concat approach on the Thin Obs. & Sph.  environment with 10 new seeds. We compare the curves between our new runs with the original 5-seed runs here (https://drive.google.com/file/d/1RWwYNMaC0C2iOYe33M_vQj6-nllYpJN6/view?usp=sharing ). We can see that the results are quite similar.
> Due to the limitation of time and compute resources, we were not able to re-run all experiments with 10 seeds at this point. But we will re-run all of them with 10 seeds for future revision.
>
> **Q:** *“what exactly does the collision happened represent. Are these total number of collisions over 1000 steps? ... Why not just 'the number of collisions with obstacles per 1k step long episode or something along those lines?”*
>
> **A:** Thank you for pointing out the unclear part for the total number of collisions, we would like to clarify that: In simulation, we report the average number of steps where collisions happened over the episode of 1000 steps. In the real-world, we manually count the number of collisions for each run.
> We have revised the description to “the number of collisions with obstacles per episode (with length of 1k steps)” in the paper, we also change the table with “Distance Moved (m)” to emphasize it is measuring meters.
>
> **Q:** *“typo”*
>
> **A:** Thank you for pointing our typos out. We have fixed the mentioned typos and other typos in our updated version of the paper.

---

### Official Review · Reviewer_XXAB · 2021-11-02

**Correctness:** 2
**Technical Novelty And Significance:** 2
**Empirical Novelty And Significance:** 2
**Recommendation:** 6
**Confidence:** 4

**Main Review:**

The main strengths of the paper:

(1) Proposed a novel transformer based architecture that can train visual-locomotion policies end-to-end, and demonstrated good navigation/obstacle avoidance/uneven terrain walking results in the simulation.

(2)  Zero-shot real world transfer to a A1 robot and demonstrates walking + navigation behavior in various environments.

The main weakness of the paper:

Not enough baselines to compare with. As the authors cited, there are many approaches to tackle visual locomotion + navigation problem besides end to end training. For example in the hierarchical approach one can combine: learned/optimization based navigation + pre-trained or hand tune walking (i.e. MPC) motions.  So in total even the hierarchical approach can have four different combinations to compare with. Yet I saw non of them here. I would say the authors should include at least one or two such baselines to compare with, and document the performances and cons and pros.




**Summary Of The Paper:**

In this paper, the authors proposed a transformer based architecture that combines both visual (depth) and proprioceptive inputs (i.e. IMU and joint angles) to solve visual locomotion tasks. The authors demonstrated that their approach can solve challenging visual navigation tasks and locomotions task on uneven terrains. The proposed method out perform proprioceptive only, visual only, and HRL baselines. The sim trained policy has been demonstrated on the real A1 hardware.

**Summary Of The Review:**

The authors proposed to use transformer architecture to solve visual locomotion + navigation tasks. The proposed approach is compared with a few end-to-end trained baselines including HRL and has demonstrated advantages. The authors also deployed the trained policy successfully to the real robot.

---

> ### Author Response · Authors · 2021-11-21
> **For Reviewer XXAB**
>
> Dear reviewer, thank you for all the detailed comments and suggestions. We address your concern on the weakness of our paper as follows:
>
> **Q:** *“Not enough baselines to compare with...learned/optimization based navigation + pre-trained or hand tune walking (i.e. MPC) motions”*
>
> **A:** Please see our answer in the general comment. We provide a baseline using a high-level vision-based controller with a low-level MPC controller. Compared to our approach, we generally find the baseline performs worse in running on challenging terrains and planning on avoiding obstacles.

---

### Official Review · Reviewer_scjr · 2021-11-04

**Correctness:** 3
**Technical Novelty And Significance:** 3
**Empirical Novelty And Significance:** 3
**Recommendation:** 8
**Confidence:** 4

**Details Of Ethics Concerns:**

I'm worried about the military use of the quadruped robot:
https://www.newscientist.com/article/2293908-us-military-may-get-a-dog-like-robot-armed-with-a-sniper-rifle/#:~:text=The%20US%20military%20may%20be,Vision%20series%20of%20legged%20robots.&text=The%20robot%20is%20fitted%20with,powerful%206.5mm%20sniper%20rifle.

**Main Review:**

[Strength]

This paper tackles an important question of how to incorporate visual information in learning policies for quadrupedal locomotion, where most existing learning-based control of quadruped robots in the published works only considered proprioceptive information, and the robots are essentially "blind."

The use of visual information can allow the robots to be less conservative and plan their actions for a longer time horizon, as has been evident from the authors' comparison with a state-only baseline that only considers proprioceptive inputs.

This paper has extensive experiments and in-depth analysis in simulation, which provides a good reference for the readers to understand the benefits and limitations of different design choices.

The real-world demo from sim-to-real transfer also provides concrete empirical evidence on the practical use of the proposed method.


[Weakness]

While I like the direction this paper is going, I'm not entirely convinced that the real-world experiments in the paper fully demonstrate the necessity of visual information. For example, in [1, 2], the authors have shown working demos on terrains seemingly much more challenging than this paper. [1, 2] also showed examples of stair climbing, a task where vision is supposed to be extremely helpful: a blind robot may have to make a few failed trials before it knows the height of a stair. You could also imagine the benefit of vision in cases that require more precise footstep planning (e.g., https://youtu.be/k7s1sr4JdlI?t=176). The paper will be much stronger by including some more concrete comparisons with the current state-of-the-art learning approaches on what can be made possible via vision while previous blinds robots struggle.

While I agree that vision is important for robots to make long-term plans and autonomously traverse around obstacles, I'm not sure whether this paper's approach is better than more classic robotic pipelines. For example, instead of treating the depth image as a 2D grid and processing it using CNN, one could use the depth camera to build a 3D map of the surrounding environment and blend in the explicit notion of what's traversable and what's not. You can then plan the trajectory based on the perception results. This seems to be how Boston Dynamics' Spot uses the visual information (https://www.youtube.com/watch?v=Ve9kWX_KXus) and has shown great generalization ability in real-world scenarios — showing examples of how this paper's way of using visual information is better than classic pipelines may be essential to claim improvements.

Continuing my previous point, it would be better if the authors could include more discussion on the current state of quadrupedal locomotion both in academia and industry, where Boston Dynamics' Spot is seemingly better in terms of generalization and robustness than any of the reinforcement learning-based approaches. The authors may also shed light on under which scenarios we should choose RL-trained robots over Spot.

How does the method work in the real world if there are moving obstacles, e.g., humans and other animals?

How well does the method work compared with the built-in controller of the robot?

The paper may need a few passes of proofreading where the current manuscript includes a lot of typos, just to name a few:
Section 1, contribution bulletin points: We the propose LocoTransformer, ...
Section 3: a MDP --> an MDP
Section 6: The visual inputs also inputs the locomotion ...


[1] Joonho Lee, Jemin Hwangbo, Lorenz Wellhausen, Vladlen Koltun, Marco Hutter, "Learning Quadrupedal Locomotion over Challenging Terrain"
[2] Ashish Kumar, Zipeng Fu, Deepak Pathak, Jitendra Malik, "RMA: Rapid Motor Adaptation for Legged Robots"

=====================

[Post Rebuttal]

I thank the authors for the detailed feedback and additional experiments, which addressed most of my concerns. Great job! I have also read the reviews from other reviewers and decided to raise my score to 8: accept, good paper.

**Summary Of The Paper:**

This paper proposes to incorporate the proprioceptive and visual information together for quadrupedal locomotion. The authors introduce a new model architecture named LocoTransformer that consists of separate modality encoders for proprioceptive and visual inputs, the output of which is fed through a shared Transformer encoder to predict actions and values.

Through experiments, the authors demonstrate that the robot, with the help of both the proprioceptive and visual inputs, can walk through different sizes of obstacles and even moving obstacles. They have also transferred the learned policy from simulation to a real robot by running it indoors and in the wild with unseen obstacles and terrain.

**Summary Of The Review:**

I like the direction this paper is going: combining visual and proprioceptive information to train RL agents for quadrupedal locomotion. I also love that the authors include real-world demos of the learned policy on a physical robot. However, my main concern is that, although there are extensive evaluations in the simulation, the current set of real-world examples may not be sufficient to show the benefit of visual inputs. Examples like climbing stairs or scenarios that require more precise footstep planning would make the paper much stronger.

I'm generally excited about the progress in this direction, thus I'm currently leaning towards the acceptance side, but I hope the authors can address the issues mentioned above.

---

> ### Author Response · Authors · 2021-11-21
> **For Reviewer scjr**
>
> Dear reviewer, thank you for all the detailed comments and suggestions. To address your concern, we would like to provide response to your comments here:
>
> **Q:** *“While I like the direction this paper is going, I'm not entirely convinced that the real-world experiments in the paper fully demonstrate the necessity of visual information... what can be made possible via vision while previous blinds robots struggle.”*
>
> **A:** As stated in the paper, using vision provides guidance for avoiding large obstacles such as boxes and trees. As shown in this video (https://drive.google.com/file/d/10E_HOXxSTtkt9juDsqP_Ah1gkweTodkf/view?usp=sharing ), without using vision but just training the blind policy with state inputs and environment randomization, the robot will get stuck in front of the tree. Even though the robot might get through the obstacle (see the video on the right side), it needs to interact with the obstacle using its front head, which is unsafe behavior and it might do damage to the robot.
>
> **Q:** *“how this paper's way of using visual information is better than classic pipelines may be essential to claim improvements.”*
>
> **A:** Please see our answer in the general comment. We provide a baseline using a high-level vision-based controller with a low-level MPC controller. We believe this also sheds light on how RL can work better than classic pipelines.
>
> **Q:** *“discussion on the current state of quadrupedal locomotion both in academia and industry.”*
>
> **A:** We illustrate why quadrupedal locomotion research in academia is still important with three points:
>
> (i) **Demo v.s. comprehensive study.** While companies such as Boston Dynamics provide cool videos, we need to understand that these are well-made demos for commercial purposes. The cost of making such a demo video might be larger than funding a graduate student. And it is not clear the exact engineering details for the complexity of making the video. On the other hand, our paper provides a comprehensive scientific study on a new and **simple** algorithm. Our experiments quantitatively evaluate how well the proposed method performs. While it is not providing demo videos with millions of views on YouTube, our results are **reproducible** and it can be followed by other researchers.
>
> (ii) **Open-source learning pipeline.** We are committed to releasing the code and the environment in our experiments, as stated in our reproducibility statement in the paper. Since there is rarely available code online for locomotion research, especially using visual guidance. We believe that by releasing our code to the community, more researchers can start working on this problem together. On the other hand, Boston Dynamics has not open-sourced their machine learning pipeline.
>
> (iii) **Price.** The Spot robot is sold for \\$74,500, while the Unitree A1 robot we use costs \$10,000. We believe that experiments with lower-cost robots open up more opportunities and create motivations for other researchers. In fact, we see a lot of research work has been done with the Unitree A1 robot, but rarely any work has been published with the Spot robot.
>
> **In general, we believe gathering more talents at scale in solving the problem by open source and using lower-cost platforms has great potential. Fundamentally, it is the purpose of publishing papers itself.**
>
>
> **Q:** *“How does the method work in the real world if there are moving obstacles, e.g., humans and other animals?”*
>
> **A:** We deploy our policy trained on moving obstacles in simulation to the real world, and show the video here (https://drive.google.com/file/d/120ASEEu108sKSPW7Y9LV0x0TOOmvEBup/view?usp=sharing ). We experiment with three different scenarios: 1) Objects seen during training moving around 2) Unseen objects coming from the side 3) Unseen objects moving towards the robot. Our approach is able to generalize to avoid moving humans and moving objects. This is also a potential advantage of learning end-to-end RL policy over classic methods since re-planning with dynamic objects around can be computationally heavy.
>
> **Q:** *“How well does the method work compared with the built-in controller of the robot?”*
>
> **A:** Please see our general comment, where the built-in controller is applied.
>
> **Q:** *“typos”*
>
> **A:** Thank you for pointing it out. We have fixed the mentioned typos and other typos in our updated version of the paper.

---

### Author Response · Authors · 2021-11-21
**General Response and Revision Summary**

Dear Reviewers, we appreciate all the detailed comments and helpful suggestions. We have highlighted the changes in red in the revised version of our paper (except for fixing typos). Here we provide a general comment on a new baseline and also an overview of our changes.

**Comparison with a classical robotics pipeline:**

We conducted experiments to compare the performance of our method and a baseline that combines a high-level vision-based controller with a low-level default controller (i.e., a more classical way). Specifically, we train a vision-only high-level controller that provides the forward speed and angular speed for a low-level MPC controller. Please see Appendix D in our revised paper for more implementation details for this baseline.  We find there are three major drawbacks using this approach:

(i) It is hard for the MPC controller to robustly handle challenge terrains. In this video (https://drive.google.com/file/d/1yZ6pB3Cjdg7w_AAbkE6dR6rT8OCNjDc4/view?usp=sharing ), we show that the robot falls (although later stands back) while walking in the forest using the MPC controller. Such behavior could be harmful to the robot. On the other hand, our method is able to adapt to challenging terrains and walk smoothly.

(ii) When facing obstacles like trees, such as in this video (https://drive.google.com/file/d/1W_kAbtZeA0uGs6Q5CgJAhA63eR82AyqY/view?usp=sharing ), the baseline cannot plan well with the MPC controller for avoidance. It tries to avoid the tree, but since the motion is not very flexible and the high-level transitions are not smooth, it collides with the side of the tree and bounces away to come across the obstacle. While the agent is still moving forward, having more collisions makes the robot unsafe. On the other hand, when trained end-to-end with both vision and legged motion, our approach can flexibly change from a moving forward motion to a turning motion and adjust the speed at the same time. We generally observe much more **diverse motions** emerge and our policy is able to **transition smoothly** between these motions according to vision.

We also provide quantitative comparisons between these two approaches. Similar to what we observed from the videos, the baseline will create more collisions even though this helps walking longer distances.

| Methods            | Distance Moved ↑ | Collision Times  ↓ |
| ------------------ |:----------------:| :---------------:|
| Vision + Controller   | 11.6  ± 3.5       | 1.9 ± 0.8       |
| Our LocoTransformer     | 9.6 ± 2.0       | 0.0 ± 0.0       |

For in-door environments where the obstacles are larger, there is less chance for the baseline to avoid the obstacle when the robot is about to collide into it. Thus the baseline performs worse in both collision times and distance moved compared to our approach.

| Methods            | Distance Moved ↑ | Collision Times  ↓ |
| ------------------ |:----------------:| :---------------:|
| Vision + Controller   | 7.5  ± 1.5       | 3.4 ± 0.8       |
| Our LocoTransformer   | 9.6 ± 2.2       | 0.3 ± 0.5       |

(iii) Extra computation resources are required with the baseline. We deploy the default MPC controller with the Unitree A1 robot so the planning can be conducted on-board. For other customized controllers, such as the MPC controller used by MIT cheetah 3 [1], the inference speed is at around 120Hz on the Unitree A1 robot. To overcome the limitation of computation, an external laptop is used in [2,3]. In contrast, our end-2-end learned policy is able to utilize the on-board GPU to perform real-time inference.

[1] Carlo et al. ”Dynamic Locomotion in the MIT Cheetah 3 Through Convex Model-Predictive Control”

[2] Yang et al. “Fast and Efficient Locomotion via Learned Gait Transitions”

[3] Sun et al. “Online Learning of Unknown Dynamics for Model-Based Controllers in Legged Locomotion”

**List of changes in the paper:**
- Add a section Appendix D to include implementation and results of the new baseline with visual navigation + classical controller.
- Add more details on the implementation of our methods in Appendix B, including hyperparameter details and randomization details.
- Add unit (Meter) for the distance moved metric in all tables.
- rephrase the description for the number of collisions in the evaluation metric.

---

### Decision · Program_Chairs · 2022-01-20

**Decision:**

Accept (Spotlight)

**Comment:**

The paper addresses vision-based and proprioception-based policies for learning quadrupedal locomotion, using simulation and real-robot experiments with the A1 robot dog. The reviewers agree on the significance of the algorithmic, simulation, and real-world results. Given that there are also real-robot evaluations, and an interesting sim-to-real transfer, the paper appears to be an important acceptance to ICLR.